# Development and Modification of Pre-miRNAs with a FRET Dye Pair for the Intracellular Visualization of Processing Intermediates That Are Generated in Cells

**DOI:** 10.3390/s21051785

**Published:** 2021-03-04

**Authors:** Yukiko Kamiya, Hiroshi Kamimoto, Hongyu Zhu, Hiroyuki Asanuma

**Affiliations:** Department of Biomolecular Engineering, Graduate School of Engineering, Nagoya University, Furo-cho, Chikusa-ku, Nagoya 464-8603, Japan; kamimo0819@gmail.com (H.K.); zhu.hongyu@i.mbox.nagoya-u.ac.jp (H.Z.)

**Keywords:** pre-miRNA, miRNA, FRET, RNAi, imaging, nucleic acid drug

## Abstract

microRNAs (miRNAs) are small non-coding ribonucleic acids (RNAs), which regulate gene expression via the RNA interference (RNAi) system. miRNAs have attracted enormous interest because of their biological significance and disease relationship. In cell systems, the generation of miRNA is regulated by multiple steps: the transfer of primary miRNA from the nucleus to the cytosol, the generation of the precursor-miRNA (pre-miRNA), the production of double-stranded RNA from pre-miRNA by the Dicer, the interaction with protein argonaute-2 (AGO2), and the subsequent release of one strand to form miRISC with AGO2. In this study, we attempt to visualize the intermediates that were generated in the miRNA-maturation step in the cells to acquire a detailed understanding of the maturation process of miRNA. To achieve this, we developed pre-miRNAs labeling with a Dicer- or AGO2-responsible fluorescence resonance energy transfer (FRET) dye pair. We observed that modifications with the dye at suitable positions did not interfere with the biological activities of pre-miRNAs. Further, imaging analyses employing these pre-miRNAs demonstrated that the processing of pre-miRNA promoted the accumulation of miRNA at the specific foci in the cytosol. The FRET-labeled pre-miRNA would further elucidate the mechanisms of the RNAi process and provide the basis for development of nucleic acid drugs working in the RNAi system.

## 1. Introduction

RNA interference (RNAi) is an endogenous system, which downregulates gene expression in a post-transcriptional manner. microRNAs (miRNAs) act as regulators of the RNAi system by binding with target messenger RNAs (mRNAs) [1,2,3]. The aberrant expression and function of certain miRNAs are linked to diseases [4,5,6,7]. Therefore, the development of an artificial miRNA or miRNA inhibitor has been considered in the nucleic acid therapeutic field [8,9,10,11]. In cells, the generation of miRNAs is properly controlled by multiple scission steps [1,2,3]. Primary miRNAs (pri-miRNAs), which are the precursors of miRNA, are long RNAs possessing a local hairpin structure. pri-miRNAs are processed by Drosha to yield ~70 nucleotide-long products that are called pre-miRNAs. The loop regions of pre-miRNAs are eliminated by an endonuclease Dicer, and the resultant double-stranded RNAs (dsRNA, miRNA/miRNA*s) interact with the Argonaute (AGO) protein to produce pre-miRISC. The subsequent removal of miRNA* from the miRNA/miRNA* complex generates miRISC, and the residual miRNA can be hybridized by complementary mRNA to negatively regulate a target gene. Thus, the maturation of miRNA requires processing by endo-type ribonucleases. Interestingly, both the molecular state and intracellular distribution patterns are changed in this process [2,12]. Upon processing pri-miRNA by Drosha, the resultant pre-miRNAs are translocated from the nucleus to the cytosol. It is known that active miRISC is accumulated in the processing body (P-Body) where the decapping, deadenylation, and degradation reactions of mRNA, which was captured by miRISC, occur [13,14,15,16]. Importantly, the miRNA biogenesis pathway is dysregulated by genetic and epigenetic alteration in disease cells such as cancer [7].

To further investigate the maturation of miRNA, the activation of the miRNA mimic and the interaction of anti-miRNA oligonucleotide with miRNA in this system, the imaging studies of the related proteins, the processing of the miRNA intermediates, and miRNA are beneficial. The intracellular distributions of the RISC proteins, including Dicer and AGO2, and the P-body localizing proteins, such as GW182, Dcp1a, and Dcp2, have been well-analyzed by immunofluorescence methods or fluorescent protein-based protein imaging [13,14,15]. Conversely, the visualizations of miRNA and its precursors remain a challenge. The fluorescent labeling technique is one of the methods of visualizing miRNAs or short interfering RNA (siRNA) working in the RNAi system of cells. Although fluorescent in situ hybridization or molecular beacon-based analyses employing fluorescently labeled complementary antisense can detect the target miRNA [17,18,19], it would inhibit the function of the target RNA. Therefore, the development of the direct labeling of miRNA or siRNA with fluorophores has been considered. The uptake by cells and the distribution of siRNA or miRNA in the cytosol after endosomal escape were detected by a modified dye [20,21,22]. A single molecular imaging reveals that miRNA is associated with a P-body [23]. For further details on miRNA or siRNA, the intracellular trafficking of the double-strand (ds) state was analyzed by fluorescence resonance energy transfer (FRET) dye pairs [24,25,26]. Further, the formation of mature RISC was demonstrated with siRNA that was modified with a fluorophore–quencher dye pair [27]. Regarding pre-miRNA, the quencher and fluorophore labeling of both terminals of the stem–loop structure of pre-miRNA were proposed as a tool for the in vitro detection of Dicer activity [28]. However, such modifications of the terminal with dyes are not suitable for the imaging analyses because the terminal regions are cleaved by the Dicer, and the resultant dye-conjugated short RNA fragments are released from pre-miRNA in the cell. Therefore, to monitor the processing intermediates that are generated in the miRNA maturation step, the modifications of the dyes in pre-miRNA are required. Although the hetero-labeling of functional groups on RNA utilizing click-based methods have been attempted, there are still setbacks, including low yields and complicated preparation steps [29,30,31]. Here, we developed FRET-type pre-miRNAs by the D-threoninol technology [32,33]. We designed Dicer-responding pre-miRNA (miLoop) and AGO-responding pre-miRNA (miStem) (Scheme 1). In miLoop, donor and acceptor dyes were introduced into the loop and stem regions, respectively. However, in the miStem case, the donor and acceptor dyes were introduced into the 3′ and 5′ stem regions, respectively. The FRET signal of miLoop is expected to be canceled by the cleavage of the loop region by the Dicer, although that of miStem is expected to be unchanged. Further, the FRET from miStem can be monitored in the pre-miRNA and miRNA/miRNA* states, although it would be canceled after unwinding the duplex RNA in miRISC. To achieve these designs, we optimized the positions of the FRET dyes on pre-miRNAs without negatively impacting the reaction of Dicer and RNAi activity. The data indicated that the processing products of pre-miRNAs, i.e., miRNA/miRNA* and miRNA, could be monitored in the cells. Pre-miRNAs can be applied to investigate the RNAi system and develop nucleic acid drugs, such as miRNA mimics and anti-miRNA oligonucleotides.

## 2. Materials and Methods

### 2.1. Syntheses of the Modified RNAs

Oligonucleotides containing only natural bases were obtained from Hokkaido System Science Co., Ltd., Sapporo, Japan or Fasmac, Kanagawa, Japan. For the preparation of dye-modified RNA, phosphoramidite and D-threoninol monomers of Perylene, TO, and Cy3 were synthesized according to a previous report with minor modifications [34,35,36]. The synthesis of BO-conjugated phosphoramidite monomer is described in the Appendix A. The modified dsRNAs, the 5′ fragments of pre-miR17 and the 3′ fragments of pre-miR17 were synthesized utilizing an ABI-3400 DNA synthesizer (Applied Biosystems, Foster City, CA, USA) with conventional and dye-carrying phosphoramidite monomers. After the recommended workup, the synthesized RNAs were purified by high-performance liquid chromatography in a reverse-phase column, and their identities were confirmed by matrix-assisted laser desorption ionization time-of-flight mass spectrometry (MALDI-TOF MS) analyses utilizing Autoflex II (Bruker Daltonics, Billerica, MA, USA). The MS data of the synthesized RNAs are summarized in the Appendix A.

### 2.2. Preparation of the Modified Pre-miRNAs

To prepare the fluorescently modified pre-miR17s, the 5′ and 3′ segments of RNAs were enzymatically ligated. The ligation reaction was performed for 16 h at 16 °C for employing a T4 RNA ligase (TAKARA Bio Inc., Kusatsu, Shiga, Japan). The product was separated by 15% acrylamide gel containing 8 M urea. The ligated products were extracted from the gels by water and purified via ethanol precipitation. The purified pre-miR17s were monitored by 15% acrylamide gel containing 8 M urea (Appendix A).

### 2.3. Measurements of UV–VIS Spectra and T_m_

The modified dsRNA (2.0 μM for RNA possessing acceptor fluorophore and 2.4 μM for RNA possessing donor fluorophore) or pre-miR17 (2.0 μM) were dissolved in 10 mM sodium phosphate buffer (pH 7.0) with 100 mM NaCl. The melting curves were obtained with a Shimazu UV-1800 (Shimazu, Kyoto, Japan) equipped with a programmable temperature controller using 10-mm quartz cell. Melting curves were measured with the change in absorbance at 260 nm versus temperature (1.0 °C∙min^−1^). *T*_m_ values were determined from the maximum in the first derivative of the melting curve.

### 2.4. Fluorescent Spectroscopic Measurements

The fluorescence measurements of RNAs were performed on a JASCO model FP-6500 (JASCO Inc., Tokyo, Japan) at 20 °C. For dsRNA, 1.0 µM RNA possessing a donor fluorophore, 1.2 µM RNA possessing an acceptor fluorophore were dissolved in the buffer, which was composed of 10 mM sodium phosphate and 100 mM NaCl at pH 7.0. For the pre-miR17, 1.0 µM RNA was dissolved in the buffer. The fluorescence spectra of oligonucleotides were recorded upon excitation at 430 nm for S7Per/A5TO and S7Per/A5Cy3 or 458 nm for S7BO/A5TO, S7BO/A5Cy3, miLoop, and miStem in the medium gain setting. Fluorescence intensities are given in arbitrary units (a.u.).

### 2.5. Dicing Assay

Pre-miRNA (6.0 µM) was incubated with Dicer (0.1 U/µL, Genlantis Inc., San Diego, CA, USA) in a Dicer reaction buffer, which was supplemented with 1.0 and 2.5 mM ATP and MgCl_2_, respectively. The aliquots that were removed at desired times were added to load the buffer containing ethylenediamine tetraacetic acid (EDTA, 50 mM) and bromophenol blue (0.025%), after which they were subjected to electrophoresis in 20% polyacrylamide gel containing 10% glycerol for 2 h at 750 CV. native-pre-miR-17 was stained by SYBR^®^ Gold (Thermo Fisher, Waltham, MA, USA). The gels were analyzed with a Typhoon FLA 9500 (GE Healthcare, Chicago, IL, USA).

### 2.6. Cell Culture

The HeLa cells were cultured in a Dulbecco’s Modified Eagle Medium (DMEM), which was supplemented with 10% fetal bovine serum, 80 µg/mL penicillin, and 90 µg/mL streptomycin. The cells were cultured with 5% CO_2_ in humidified air at 37 °C.

### 2.7. Luciferase Assay

For the dual-luciferase assays, the reporter plasmid, pmirGLO-miR17, was prepared. In pmirGLO-miR17, the miR-17 binding sequence (5′-CTACCTGCACTGTAAGCACTTTG-3′) was inserted between the *Nhe I* and *Sal I* sites in the 3′-UTR region of the gene encoding the firefly luciferase in pmirGLO (Promega, Madison, WI, USA). The co-transfections of the HeLa cells with pre-miR17 (final concentrations = 30 nM) and 100 ng of pmirGLO or pmirGLO-miR17 were performed with Lipofectamine^TM^ 2000 (Invitrogen, Carlsbad, CA, USA) in 96-well plates according to the manufacturer’s instructions. After 24 h of incubation at 37 °C, 75 µL of the medium was removed, and 75 µL of Dual-Glo^®^ reagent (Promega) was added. The firefly luciferase luminescence was measured on a Multi-label Plate Reader (EnSpire^TM^, Perkin Elmer, Waltham, MA, USA). Subsequently, 75 µL of Dual-Glo^®^ Stop and Glo^®^ reagent was added, and the *Renilla* luciferase luminescence was measured.

### 2.8. Fluorescent Imaging Analyses of the Pre-miRNA Transfected HeLa Cells 

The HeLa cells (70,000 cells/mL) were split on the bottom of a 35 mm dish in a 2 mL medium and cultured overnight. For the lipofection, the transfections of the HeLa cells with the modified pre-miR17 (final concentrations = 30 nM) were performed with Lipofectamine^TM^ 2000 (Invitrogen) in 96-well plates according to the manufacturer’s instructions and incubated for 12 h. For the methods of loading the beads, Opti-MEM containing the modified pre-miR17 (2 μM) was added to the dish after removing the medium, and the cells were incubated for 2 min. The glass beads (2 mg, <106 µm, acid washed, Sigma Aldrich, St. Louis, MO, USA) were added to the medium, after which the dish was shaken for 30 s. The HeLa cells were incubated for 1 h at 37 °C. The cells were washed with Opti-MEM, and the medium was changed to DMEM without phenol red. 

For immunofluorescent analyses, the HeLa cells were fixed in phosphate-buffered saline (PBS) containing 4% paraformaldehyde and treated with PBS containing 0.2% Triton-X100 for 5 min at RT. Subsequently, the cells were stained with anti-AGO2 (FUJIFILM Wako, Osaka, Japan) and Alexa Fluor 648-conjugated anti-mouse IgG antibodies (Abcam, Cambridge, UK), anti-GW182 (Santa Cruz Biotechnology, Dallas, TX, USA) and Alexa Fluor 488-conjugated anti-goat IgG antibodies (Invitrogen, Carlsbad, CA, USA), or anti-Dicer (Abcam) and Alexa Fluor 647-conjugated anti-mouse IgG antibodies (Abcam). For time course analysis, after introduction of the miStem into the cell by use of the beads loading method, images of the cells were obtained at specific time point. The HeLa cells were visualized on an FV-1000 or FV-3000 confocal laser microscopy (Olympus, Tokyo, Japan). The fluorescent signals were recorded as follows: Cy3: Ex = 543 nm, Em = 555–625 nm; Alexa647: Ex = 633 nm, Em 645–745 nm; FRET: Ex = 458 nm, Em = 555–625 nm. The fluorescent images were processed with ImageJ. 

## 3. Results

### 3.1. Activities of Pre-miR17s That Were Modified with the Fluorophore at Various Positions

As a model pre-miRNA, pre-miR17 was utilized in this study. To introduce the fluorescent dye into pre-miR17, we first determined the appropriate positions to be modified on pre-miR17 utilizing pre-miR17s that were modified with perylene (Per) through the D-threoninol linker at various positions (Figure 1a). The fluorophore-modified pre-miR17s were successfully synthesized by the ligation of the 5′ and 3′ termini RNA segment of the oligonucleotides (Appendix A). The RNAi activities of pre-miR17s that were modified by Per through D-threoninol were evaluated by luciferase assays. For this assay, a pmirGLO-17 plasmid possessing an miR-17-5p target site in the 3′-UTR region of the luciferase coding region was utilized. 

Compared with the cells that were transfected with mirGLO, which does not possess an miR-17 target site, the luciferase expression in pmirGLO-17, which transfected the HeLa cells, was reduced by the endogenous miR-17 (Figure 1b). The co-transfections of native pre-miR17 and pmirGLO-17 in the cells further reduced the luciferase expression. Interestingly, the reduction levels in luciferase by the Per-modified pre-miR17s depended on the position where Per was introduced. We observed that pre-miR17s with Per on the 5′ side of the stem region exhibited lowered RNAi activities except when the modification was at position 4. Conversely, pre-miR17 that was modified with Per in the loop region and 3′ side of the stem region exhibited sufficient RNAi activities, which were of a similar degree to those of native pre-miR17. Positions 1–3 are located in the seed region of miR-17, and position 5 was the interaction site with a PAZ domain of AGO2 [37,38,39], suggesting that the modifications at these sites interfered with the correct result of RISC formation during the reduction of the RNAi activities. Thus, we observed suitable modification sites on pre-miR17, and the sites did not interfere with its activities. Numbered lists could be added, as follows:

### 3.2. Investigation of Suitable FRET Dyes That Were Introduced into Pre-miR17

Next, we surveyed the suitable FRET dye pairs that were introduced into pre-miR17. We prepared dsRNAs that were modified with Per–TO, Per–Cy3, BO–TO, and BO–Cy3 to test the FRET efficiencies (Figure 2). Each dye was introduced into the individual RNA strand through the D-threoninol backbone. At the early stage of this analysis utilizing dsRNA possessing the TO–TR pair, we observed that the acceptor fluorescence was quenched when the dye pairs were located at a complementary position on the sense and antisense strands of dsRNA (Appendix A). Therefore, the dye pairs were introduced at the 7th position from the 5′ end of the sense strand and 5th from the 3′ end of the antisense strand. The fluorescent emission spectra of S7Per/A5TO revealed that the donor and acceptor fluorescence around 480 and 520 nm, respectively, were observed, and they overlapped significantly (Figure 2c). Regarding S7Per/A5Cy3, S7BO/A5TO, and S7BO/Cy3, the donor fluorescence decreased and the acceptor remarkably increased concurrently (Figure 2c). The *T*_m_ and UV spectra of these modified dsRNA suggested that the fluorophores were intercalated into the dsRNAs (Appendix A). Especially the *T*_m_ data suggested that the Per-modified dsRNA were stabilized through stacking interactions between Per and nucleobase pair. The overlapping of the donor and acceptor fluorescence must be minimized with sufficient FRET efficiency in the imaging assay. Among the FRET dye pairs, we selected BO–Cy3 as a FRET dye pair for pre-miRNA imaging in the cells.

We prepared the BO–Cy3-introduced pre-miR17 (Figure 3a and Appendix A). Regarding miLoop, BO was modified in the loop region at position 10, and Cy3 was inserted in the 5′ stem region, at position 4. Regarding miStem, BO and Cy3 were inserted in the stem region at positions 8 and 4, respectively. *T*_m_s value of miLoop and miStem indicated that fluorphore modifications little impact on the stabilities of the stem-loop structure (Appendix A). We evaluated the fluorescence properties of these modified pre-miR17 (Figure 3b,c). The fluorescent spectra exhibited a high FRET signal around 570 nm. The reduced fluorescence of BO around 480 nm from miStem that was excited at 458 nm was observed as expected. The FRET signal around 570 nm from miLoop that was excited at 458 nm was sufficient, although there was a distance between Cy3 and BO. Based on these results, we employed these FRET-modified pre-miR17s for further biological analyses.

### 3.3. Biological Activities of the BO–Cy3 Pair-Modified Pre-miR17s

Generally, in the miRNA-generation step, the loop region of pre-miRNA is cleaved by the Dicer to produce dsRNA, which is recognized afterward by AGO2. To analyze the biological properties of FRET-modified pre-miR17, we determined whether pre-miRNAs that were recognized by the Dicer and loop regions were cleaved. pre-miRNAs were incubated with the Dicer at 37 °C, and the samples were analyzed by polyacrylamide gel electrophoresis (PAGE), and native pre-miR17 was mostly diced within 6 h (Figure 4a). Despite the slow dicing of miLoop and miStem compared with the native pre-miR17, these modified pre-miRNAs were mostly diced within 24 h (Figure 4a). These data indicate that the FRET dye-modified pre-miR17s could be recognized and cleaved by the Dicer. 

Further, we investigated whether miLoop or miStem exhibited RNAi activities in the cells via the luciferase reporter assay. In the untreated cell, the luciferase expression was suppressed by the endogenous miR-17 (Figure 4b). After the transfection of the modified pre-miR17, the luciferase expression was further reduced compared with the HeLa cell without the transfection of pre-miR17. This indicated that the modified pre-miR17s could knock down the target luciferase in the cell (Figure 4b). The data indicate that miLoop and miStem have similar activities in the cells.

### 3.4. Cell Imaging Analyses of Fluorescently Modified Pre-miR17s

To observe the intracellular localization of the fluorophore-modified pre-miR-17, the imaging analyses of the cells that were transfected with the fluorophore-modified pre-miR17 were performed. Firstly, we analyzed the transfected cells via pre-miR17 modification with Cy3 (miCy3) at position 4. We expected a fluorescence signal from the originally introduced pre-miRNA in any state, i.e., pre-miRNA, miRNA/miRNA*, or miRNA. For the transfection of pre-miR17 in the cell, the lipofection and bead-loading methods were employed. In both cases, the fluorescent spot, which originated from Cy3 was observed upon transfection in the cell (Figure 5). Regarding the lipofection, a higher number of fluorescent spots was observed compared with the beads loading method, and this implied that the lipid complex particle containing the modified pre-miR17 in the endosome and the accumulation of pre-miR17 in the cytosol were observed (Figure 5b,c). It should be also noted that we observed the fluorescent spots outside of the cell in case of the lipofection intruduction. This suggests that the lipid-pre-miR17 complex which did not enter the cells or excluded from cells were accumulated. We expected that the maturation product of miCy3 was complexed with AGO2 to form RISC in the RNAi system. Therefore, we visualized the AGO2 protein in the miCy3-transfected cells by the immunostaining method. AGO2 was distributed in the cytosol, while part of AGO2 was accumulated as foci, which correspond to the P-body non-membrane subcellular region, where RISC facilitated the RNAi reaction. In the miCy3-lipofected cells, part of the Cy3 spots was co-localized with the AGO2 foci, indicating that the processing product of miCy3 formed RISC in the cell (Figure 5b). Similarly, the cells, which transfected miCy3 by the beads-loading method revealed the Cy3 spots that were co-localized with part of the AGO2 foci (Figure 5c). The data indicated that the fluorescently modified pre-miRNA could be recognized by the RNAi-related proteins to form miRISC in the cells.

Thereafter, we performed the imaging analyses of miLoop and miStem that were transfected by the beads loading method. The specific foci of Cy3 fluorescence upon exciting Cy3 was observed in the cells (Figure 6a). Immunofluorescent labeling of the Dicer indicated that the Dicer was distributed in the cytoplasmic region and did not co-localize with the Cy3 fluorescence at specific foci (Appendix A). Conversely, the Cy3 fluorescence at the foci was co-localized with AGO2 (Figure 6a). In the imaging analyses of the miLoop-transfected cells, the BO–Cy3 FRET signals of miLoop were not detected. These data suggested that pre-miRNA was distributed in cytosol, after which it was accumulated in the foci after dicing the product from pre-miR17. Regarding miStem, although the accumulation of Cy3 fluorescence upon Cy3 excitation was observed, it was not co-localized with the Dicer, as was the miLoop case (Appendix A). The foci of Cy3 fluorescence were co-localized with AGO2 (Figure 6b). Additionally, the BO to Cy3 FRET signal, which originated from the miStem-transfected cells, was observed at the part of the foci where the Cy3 and AGO2 signals were co-localized (Figure 6b). These data suggested that miR-17/miR-17* of the dicing product from pre-miR17 and miR-17-5p in RISC were localized in the specific foci where AGO2 was accumulated. Notably, in the fluorescent images of the cells that were transfected with miLoop or miStem by the lipofection method, the lipid complex particle-containing pre-miR17 were so many that they could not be easily analyzed (Appendix A). However, similar results were obtained for the cells that were transfected with miLoop or miStem by the lipofection methods. Furthermore, we attempted to track the time course of miStem in the living cell. After introduction of miStem, distributed FRET signals from miStem were observed and then the signals were gradually decreased (Figure 7). We observed newly generated focis during the time course analysis, suggesting that miRNA/miRNA* accumulated in the specific region in cytsol. Thus, we successfully monitored the maturation of the processing intermediates that were generated from pre-miRNA to miRNA and incorporated into RISC in the cells.

## 4. Conclusions

In this study, we developed functionally active FRET-modified pre-miRNAs. We showed that the modified pre-miRNAs, which were inserted with the FRET dyes through the D-threoninol linker at suitable positions, exhibited activities that were similar to those of pre-miRNA. The findings enabled the design of pre-miRNAs that were modified with donor and acceptor dye conjugates in the inner region of pre-miRNA for FRET-based imaging analyses. The relationship between generation steps of miRNA and their intracellular dynamics have been little known yet. In our design, the dissociation of the dye was controlled by appropriate timing, i.e., the cleavage by the Dicer or the exit of one RNA strand from RISC during the maturation process from pre-miRNA to miRNA. Therefore, the molecular states of pre-miRNA, miRNA/miRNA*, and miRNA could be differentiated. Both FRET and direct fluorescent were observed in the cell imaging analyses. Employing 

FRET-pre-miRNAs, miRNA/miRNA* and miRNA were accumulated in the same specific foci as AGO2. Our analyses indicated that the production of miRNA/miRNA* from pre-miRNA by the Dicer and the subsequent association with AGO2 is the timing that alters the subcellular localization of the processing intermediates of pre-miRNA. 

Generation of miRNAs are regulated though the several ways [2,3]. In cytosol, it is known that Dicer-mediated processing is controlled by the specific binding protein for the individual pre-miRNAs, such as Lin28 and pre-let7 [40], and by RNA editing system [41,42]. Stabilities of pre-miRNAs are also controlled by endonucleases [2,3]. For example, IRE1α cleaves specific pre-miRNAs including pre-miR17 under endoplasmic reticulum stress condition [43]. In addition, non-canonical pathways of miRNA biogenesis are existed. Although precursor of miRNAs are encoded intergenic region in most cases, some of miRNAs are encoded in introns [2,3,44,45]. The intronic pre-miRNAs are released through the dicer-independent pathway. Thus, there are alternative pathways of miRNA maturation system to be solved although the general and canonical pathway of the miRNA biogenesis have been well known. Our new pre-miRNAs design could be applied for any pre-miRNAs, therefore, our system would be a beneficial tool to elucidate the mechanisms or determination of intracellular fate for individual pre-miRNAs.

In this study we used lipofection and beads loading methods for introduction of the FRET-modified pre-miR17. Although these methods can easily introduce oligonucleotides into cells, they have issues. In case of the lipofection, the large number of complexes between lipofection reagent and RNAs disturbed the imaging analyses. Due to the low efficiencies of introduction of oligonucleotides into cells by the beads loading methods, it took a lot of effort to acquire the fluorescent image of the cells. An alternative scheme is required for more comfortable imaging analyses of oligonucleotides.

In the future, the detailed imaging analyses of individual pre-miRNAs, their processing intermediates, and miRNAs under the normal and specific intracellular conditions would further elucidate the mechanisms of the RNAi system and the intracellular action of an miRNA mimic and anti-miRNA oligonucleotide drug working in the RNAi system.

## Data Availability

Data sharing not applicable.

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
