# Peer review of "Development and Modification of Pre-miRNAs with a FRET Dye Pair for the Intracellular Visualization of Processing Intermediates That Are Generated in Cells"

_sensors, 2021, doi:10.3390/s21051785_

Round 1

Reviewer 1 Report

The authors used a  D-threoninol linker as a technique for modifying pre-miRNA.
They carried out fluorescence assays of the modified miRNAs (multiple miRNA modified in different positions), FRET experiments and whether miLoop or miStem exhibited RNAi activities in the cells via the luciferase reporter assay.
They carried out biological analyses for testing the appropriate miRNA processing and maturation machinery by using a modified pre-miR17, as a model system.
This is a cutting-edge technique for modifying pre-miRNA useful for tracking the functionality of modified miRNA and several applications rather than RNA interference.
My humble opinion, some revisions are necessary before publication of this interesting manuscript.
Figure 1 Please evaluate appropriate statistical test for significance assessment across groups or pairwise comparisons of main interest.
As such, also in Figure 4 Biological properties of the FRET dye-conjugated pre-miR17s. Although it is clear that no strong differences are present
between miLoop or miStem, any statistical test is missiing and this shoud be included.
Did you verify the percentage of transfection with the fluorophore-modified pre-miR-17 and unmodified one?
Notably, did you evaluate a microscopy time course for tracking the fluorophore-modified pre-miR-17 cell localization?
Figure S1, verify in the legend "phosphor group".
Table S1, please add units and explain the differences of the A5TO and A5Cy3 MALDI-TOF-MS observed and calculated value differences.
Finally, I think the discussion could be improved with more critical limitations and future prospetcives of this study.

Author Response

Thank you for your valuable comments and suggestions for our manuscript.

According to your suggestions, we conducted several experiments and revised the text. Corrections are highlighted with yellow background in the revised manuscript. Our response is summarized in the separately attached PDF file.

We take this opportunity to express our gratitude for your constructive and useful insights.

We hope that the revised manuscript is now acceptable for publication in Sensors.

Reviewer 2 Report

The paper entitled: “Development and modification of pre-miRNAs with a FRET 2 dye pair for the intracellular visualization of processing inter-3 mediates that are generated in cells” from Yukiko Kamiya et. al. has been revised. The manuscript the development of a  pre-miRNAs 19 labeling with a Dicer- or AGO2-responsible fluorescence resonance energy transfer (FRET) dye pair as a new tool for imaging analysis. For this purpose, the authors employed several microscopy and spectroscopic techniques such as fluorescent spectroscopic measurements and confocal laser microscopy together with cell culture and luciferase assays. The paper is well-written and the finding are interesting and sound well. The conclusion are well supported by the experimental data. In my opinion this work is publishable in Sensor after minor revision has noted.

Minor points:

-More details should be added in the material and methods of the paper (section 2.3) about fluorescence measurement conditions. Please, specify the excitation wavelength used for obtaining fluorescence spectra in each case, voltage and reproducibility of the samples.

-page 6, figure 2. The colour of the spectra in Figure 2 cannot be appropriately distinguished. Please modify the layout of the figures to easily visualized the differences founded between fluorescence spectra of the free dye and the dsRNA/dye.

-The mode and the strength of interaction between dsRNA and Per, BO, TO, and Cy3 fluorescence dyes is not appropriately discussed in the whole paper. To gain insight about the difference founded among different donor/acceptor pairs, the nature of the interaction between the dye and the biopolymer must be clarified in each case.

-page 9, lines 318-319. A comment about the existence of some spots out of the HeLa cells in Figure 5b seems to be pertinent.

Author Response

(The authors gave the same response as above.)

Reviewer 3 Report

Comments for sensors-1103958

Development and modification of pre-miRNAs with a FRET dye pair for the intracellular visualization of processing intermediates that are generated in cells

Kamiya, Y., Kamimoto, H., Hongyu, Z. and Asanuma, H.

In this manuscript, the authors attempt to visualize the miRNA intermediates that were generated in the maturation step in the cells. They investigated pre-miRNA labeling with a Dicer- or AGO2-responsible FRET dye pair. They found the suitable modification position and dye for their analyses and found that modifications did not interfere miRNA function. Lastly the authors identified that the processing of pre-miRNA promoted the accumulation of miRNA at the specific foci in the cytosol.

The system the authors established is quite interesting and fascinating. However, I have some concerns and comments before accepting this manuscript.

1) The authors used only one kind of pre-miRNA, pre-miRNA17. It is important to use more than one pre-miRNA in order to claim generality.

2) It is also required to show direct binding of modified pre-miRNA to Dicer.

3) It would be nicer if the authors could discuss more specific points that they could access with their nice system.

4) Many of the pre-miRNAs are encoded in introns of other genes. It would be wonderful if the authors could discuss for processing of intronic pre-mRNA in Discussion section.

Author Response

(The authors gave the same response as above.)

Round 2

Reviewer 3 Report

The authors addressed my concerns except in vitro binding experiment. I will leave the decision to the editor about the requirement of this experiment.